# Atlantic Water warming increases melt below Northeast Greenland's last floating ice tongue

Claudia Wekerle [1] ✉, Rebecca McPherson [1], Wilken-Jon von Appen[1], Qiang Wang [1], Ralph Timmermann[1], Patrick Scholz[1], Sergey Danilov [1,2], Qi Shu [3] & Torsten Kanzow [1,4]

The 79 North Glacier (79NG) features Greenland's largest floating ice tongue. Even though its extent has not changed significantly in recent years, observations have indicated a major thinning of the ice tongue from below. Both ocean warming and an increase in subglacial discharge from the ice sheet induced by atmospheric warming could increase the basal melt; however, available observations alone cannot tell which of these is the main driver. Here, we employ a global simulation which explicitly resolves the ocean circulation in the cavity with 700 m resolution to disentangle the impact of the ocean and atmosphere. We find that the interannual variability of basal melt below 79NG over the past 50 years is mainly associated with changes in the temperature of the Atlantic Intermediate Water inflow, which can be traced back across the Northeast Greenland continental shelf to the eastern Fram Strait with a lag of 3 years.

While ice shelves in Antarctica are ubiquitous extensions of the Antarctic Ice Sheet, in Greenland only three glaciers possess a floating ice tongue: the Nioghalvfjerdsfjorden Glacier (79 North Glacier, 79NG), Ryder Glacier, and Petermann Glacier. The largest of these glaciers is the 79NG in Northeast Greenland (Fig. 1a). Its ice tongue stretches over 80 km in length in a 20 km wide fjord (exceeding the internal Rossby deformation radius of 3–4 km, Supplementary Fig. 1c), which has a maximum water depth of 900 m[1] as a result of glacial over deepening. At the mouth of the fjord, the shallowest connection between the fjord and the continental shelf is a sill of 340 m depth. Moreover, islands along the fjord mouth act as pinning points. The floating ice tongue is anchored to the sides of the fjord and progressively thins from around 500 m at the grounding line to 100 m at the pinning points (Supplementary Fig. 1f). The 79NG and its neighboring glacier, the Zachariæ Isstrøm (ZI), drain the Northeast Greenland Ice Stream which covers 12% of the Greenland Ice Sheet area[2]. Its complete melt would lead to a 1.1-m global sea level rise[3]. In contrast to the ZI, which lost its floating ice tongue in the 2010s[4], the extent of the 79NG has not changed

significantly in the past decades. This is possibly due to the pinning points at the calving front of the glacier[3]. However, the ice tongue has been thinning in the past two decades[4–6].

There are several hints that the thinning of the 79NG ice tongue can be attributed to the ocean, in particular to an increase in the temperatures of the Atlantic Intermediate Water (AIW) that occupies the deeper parts of the Northeast Greenland (NEG) continental shelf[7–9]. This year-round flow of warm ( > 1°C) and salty waters through a narrow channel into the cavity is hydraulically controlled by an upstream sill, as revealed by one year of mooring measurements at the 79NG calving front[10]. The supply of AIW leads to melting at the base of the ice tongue and undercutting of grounded ice[3]. Studies based on satellite imagery[5] estimated a negative mass balance for the ice tongue of the 79NG during the time period 2011–2015, and revealed that basal melt rates exceeded the ice discharge across the grounding line.

Apart from the ocean which carries the warm AIW into the cavity of the 79NG, the atmosphere impacts the stability of the glacier as well via subglacial discharge[11]. Runoff at the ice sheet surface results from

[1]Alfred Wegener Institute, Helmholtz Centre for Polar and Marine Research, Bremerhaven, Germany. [2]Department of Mathematics and Logistics, Constuctor University, Bremen, Germany. [3]First Institute of Oceanography, Ministry of Natural Resources, Qingdao, China. [4]University of Bremen, Bremen, Germany. ✉e-mail: claudia.wekerle@awi.de

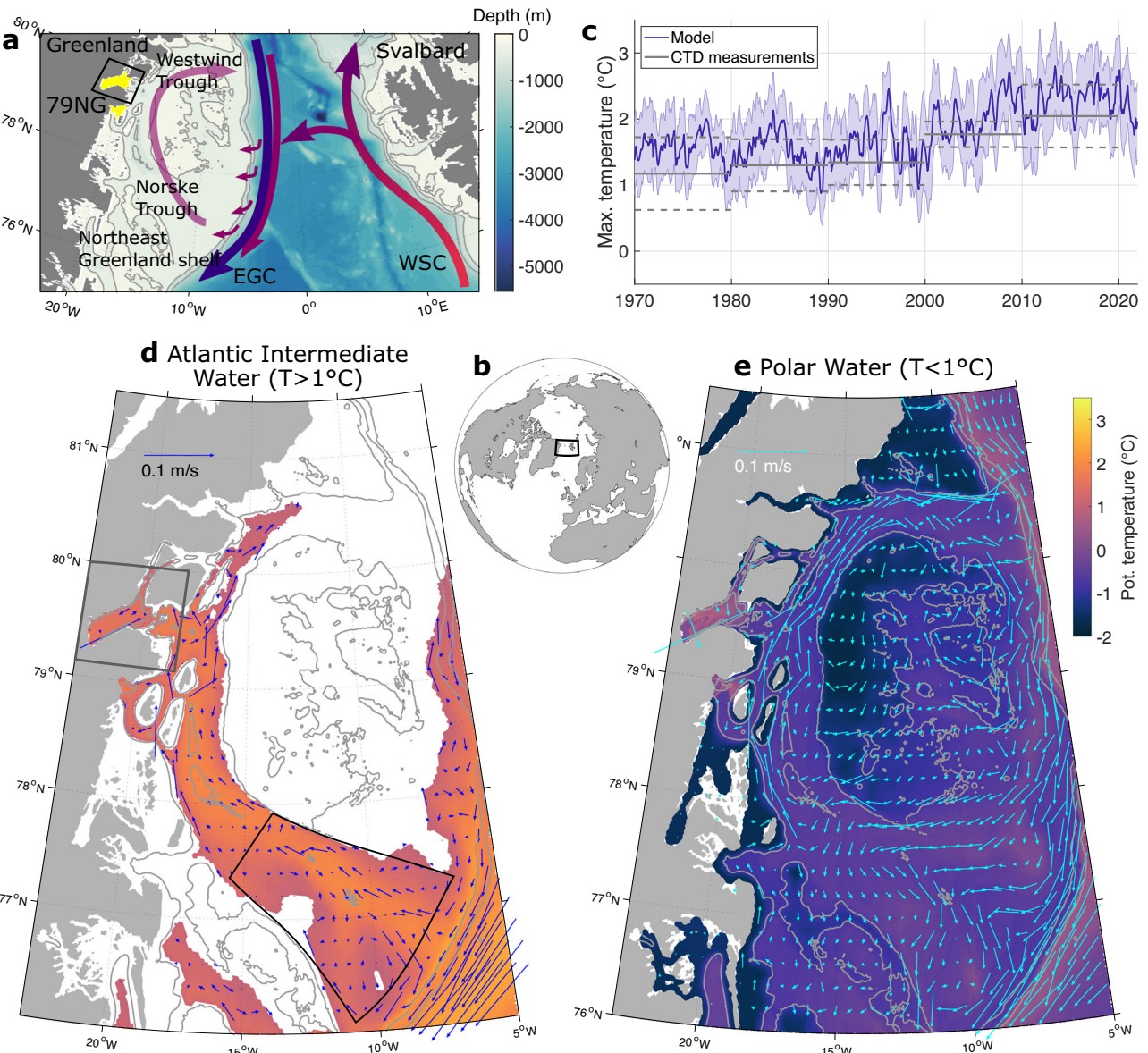

**Fig. 1 | Circulation and temperature in Fram Strait and on the Northeast Greenland (NEG) continental shelf. a** Sketch of the ocean circulation in our study region. Purple arrows indicate the West Spitsbergen Current (WSC) transporting warm and salty Atlantic Water toward the Central Arctic Ocean. The blue arrow indicates the East Greenland Current (EGC) transporting fresh and cold Polar Water southwards. On the continental shelf, warm Atlantic Intermediate Water (AIW, light purple arrow) is transported through Norske Trough towards the 79 North Glacier (79NG). The extent of the floating ice tongues of the 79NG and Zachariæ Isstrøm are highlighted in yellow. **b** Study area on a global map. **c** Monthly mean (solid line) and

associated standard deviation (shading) of simulated maximum temperature in the water column averaged over southern Norske Trough (black polygon shown in panel d). Gray solid and dashed lines indicate decadal averages and standard deviations respectively from historic hydrographic observations[45]. **d, e** Mean velocity and potential temperature averaged over the AIW layer (defined by $T > 1\,°C$) and the Polar Water layer (defined by $T < 1\,°C$), averaged over 1970–2021. In white areas, the water masses are absent. Gray lines show isobaths at 200 m, 500 m, and 1000 m. The gray box at the 79NG in **d** indicates the region shown in Fig. 2a, b.

ablation and liquid precipitation and drains to the bed of the ice sheet. It then flows towards the grounding line and, if this is below the sea surface, enters the ocean as subglacial discharge[12]. The outflowing freshwater vigorously entrains ambient waters, speeds up as a buoyant plume that develops downstream of the grounding line at the base of the ice tongue, and thereby locally increases basal melt. This fresh-water source is typically most pronounced around Greenland in summer and is almost absent otherwise. Runoff has been increasing during the past decades, a result of enhanced surface air temperatures over the Greenland ice sheet[13].

Covered by an ice tongue of several 100 m thickness, the ocean circulation in the cavity of the 79NG is thus far largely unknown.

Measurements with an Ice Tethered Mooring[14] and CTDs[15] allow a glimpse into the cavity but are constrained to single locations. Results from refs. 7,10,16,17 showed the important role that topo-graphically steered subsurface currents play in transporting the warm AIW across the continental shelf and into the cavity. Yet these continuous observations were only started in 2016, and do not allow to investigate multi-annual to decadal changes in ocean heat supply. In addition, there is yet no realistic model simulation that resolves the subsurface flow on the shelf, and that simulates the 3D circulation of AIW within the cavity where it drives the basal melt. Modeling studies exploring the ocean circulation at the 79NG and in other cavities of Greenland glaciers (e.g. Petermann and Ryder

Glaciers) have been so far restricted to idealized setups in two dimensions[18–20].

Here we present a dynamically consistent framework that connects the impact of both ocean warming and subglacial discharge with the melt rates below the ice tongue of the 79NG on interannual to decadal time scales. We apply a global 3D multi-resolution sea ice–ice shelf–ocean model, the Finite-volumE Sea ice-Ocean Model (FESOM2.1,[21]), which resolves the cavity of the 79NG with a dynamically appropriate horizontal resolution of 700 m (Methods, Supplementary Fig. 1). With this methodology we seamlessly connect the global and regional ocean circulation to the circulation in the cavity. Our simulation with realistic bathymetry and ice shelf geometry covers the time period 1970–2021. With this simulation we disentangle the drivers of the upward trend and interannual variability of the basal melt, and find that ocean warming conveyed via ocean currents has played a dominant role over the past 50 years.

## Results

### The Northeast Greenland continental shelf: ocean conditions

The 79NG is connected to the large-scale ocean circulation via a C-shaped trough system on the NEG continental shelf, consisting of the Norske Trough in the south and the Westwind Trough in the north (Fig. 1a). Originating from the North Atlantic, warm and salty Atlantic Water is transported northwards by the West Spitsbergen Current (WSC) towards the Arctic Ocean[22]. A fraction of it recirculates in the central Fram Strait[23], subducts underneath cold and fresh Polar Water (PW), and finally enters the NEG shelf via Norske Trough[7]. There, it mixes with the colder Arctic Atlantic Water (AAW) that circulates through the Arctic Ocean[24], forming AIW (defined here as waters with potential temperature $> 1\,°C$). The water column on the NEG continental shelf is thus strongly stratified, with the relatively warm and salty AIW below a layer of cold and fresh PW. Our simulation reveals a current that transports the AIW from the mouth of Norske Trough northwestwards through the trough system (Fig. 1d), in agreement with mooring measurements in central Norske Trough which revealed a 10-km-wide bottom-intensified jet carrying AIW towards the glaciers[16]. AIW is present in Norske Trough, while fading out towards Westwind Trough in agreement with historic observations[7]. This indicates that AIW is being cooled by mixing with the overlying PW on its way to the inner shelf and that it does not enter the continental shelf through Westwind Trough. The flow of PW (defined by $T < 1\,°C$, in agreement with ref. 25), forms an anti-cyclonic loop on the continental shelf. It flows northwards along the Greenlandic coast, continues eastwards through Westwind Trough, and then joins the boundary current flowing southwards along the continental slope as the East Greenland Current (EGC) (Fig. 1e).

As in-situ hydrographic observations on the NEG continental shelf are sparse due to harsh weather conditions and the year-round presence of sea ice, we compiled decadal means of hydrographic measurements in southern Norske Trough from the 1970s to the 2010s. Time-series of observed AIW temperature (vertical maximum of temperature in the water column) demonstrate a marked warming trend, reproduced by our model (Fig. 1c). Mean temperatures increased by $0.9\,°C$ in the observations and by $0.8\,°C$ in the model from the 1970s to the 2010s. Averaged over the entire time period 1970–2021, a slight positive AIW temperature bias of around $0.3\,°C$ in the mean state of the model simulation on the NEG shelf compared to the observations is found. This bias can be traced back to the Nordic Seas with a warmer-than-observed Norwegian Atlantic Current. However, the simulated AIW temperatures are mostly within the observational uncertainty range and correctly represent the increasing trend. Moreover, mooring measurements conducted in Fram Strait revealed a positive trend in the WSC temperatures of $0.5\,°C\ decade^{-1}$ for the time period 1997–2018[26], replicated by the FESOM2.1 simulation. We are thus confident that the hydrography on the NEG shelf and in the wider Fram Strait is well represented in the model, which can then be used to study the circulation in the cavity.

### Mean ocean circulation in the cavity and basal melt rates

The FESOM2.1 simulation reveals warm and salty AIW entering the cavity through its deepest channel, leaning against its northern flank (Fig. 2a, c). The inflow of AIW amounts to 69 mSv, higher than the $46 \pm 11$ mSv obtained from mooring-based estimates at the calving front[10]. The inflow occurs primarily at the main calving front and not through Dymphna Sound whose sill is too shallow for AIW to enter[17,27]. The flow of AIW into the cavity is constrained by a sill at the calving front, as shown by a transect of simulated mean temperatures following the AIW pathway from the continental shelf into the cavity (Fig. 2e). The bottom-intensified gravity current accelerates down the slope of the sill, reaching flow speeds greater than $0.3\ m\ s^{-1}$, accompanied by decreasing temperatures. This is presumably due to hydraulic control and the associated mixing and entrainment of AIW with ambient water[10,28], which reduces the thermal forcing that reaches the grounding line. This is comparable to Ryder Glacier in North Greenland, which is shielded by a bathymetric sill as well[29]. The warm AIW then circulates cyclonically around the cavity. In particular, the jet turns north, loops into the entrance of Dymphna Sound and continues along the northern slope of the cavity. On its way towards the grounding line, the jet decelerates and becomes broader (Fig. 2d). The flow is thus clearly affected by rotation, deviating strongly from a two-dimensional estuarine circulation.

The inflow of AIW results in ocean temperatures above the freshwater freezing point in the cavity throughout the year, in agreement with CTD measurements[15], which leads to strong and persistent melting at the base of the ice tongue. In the model, melting (or refreezing) at the base of the ice is computed based on the three-equation system[30] and depends on velocity-dependent salt/heat exchanges (see Methods). The simulated long-term mean basal melt rate is highest with a maximum value of $85\ m\ yr^{-1}$ close to the grounding line (Fig. 2b), where the ice tongue is thickest (Supplementary Fig. 1f). This means that the densest (and thus warmest) waters in the cavity interact with the ice there. Near the grounding line, the ice base slopes steeply upward, allowing for (and shaped by) the rising meltwater plume. The simulated spatial pattern compares well with basal melt derived from satellite imagery (Ref. 5, Supplementary Fig. 2). The mean simulated basal melt water flux amounts to $17.0 \pm 7.9\ km^3\ yr^{-1}$ (mean ± standard deviation of monthly means, equivalent to 0.54 mSv). This simulated value is higher than the remote sensing-based estimate (ref. 5, $11.9 \pm 1.6\ km^3\ yr^{-1}$), but compares well to the estimate based on heat transport into the cavity from mooring measurements (ref. 10, $17.8 \pm 5.2\ km^3\ yr^{-1}$). Note that the remote sensing based estimate may be on the lower side as it does not take into account basal melting close to the grounding line.

Both basal melt water and subglacial discharge (the latter contributing around 20% of the overall freshwater input to the cavity) sustain a meltwater plume. Driven by its buoyancy and the effect of the Coriolis force, the plume accelerates along the southern flank of the cavity, reaching mean velocities $> 0.3\ m\ s^{-1}$ at section b-b' (Fig. 2d). On its upper boundary, freshwater from basal melt is injected along the plume's path, whereas on its lower boundary, warm and saline ambient water of AIW origin is entrained. When reaching a level of neutral buoyancy, the plume detaches from the base of the ice tongue. The bulk of the submarine meltwater exits the cavity at around 150 m depth (Fig. 2c). This agrees with measured noble gas concentrations, indicative of submarine meltwater, which peak between 100 and 150 m at the 79NG calving front[31]. We find that most of the export occurs through the main calving front (mean transport of 62 mSv, 82%), whereas the export through the narrow Dymphna Sound is smaller (14 mSv, 18%). When reaching the continental shelf, the meltwater in the

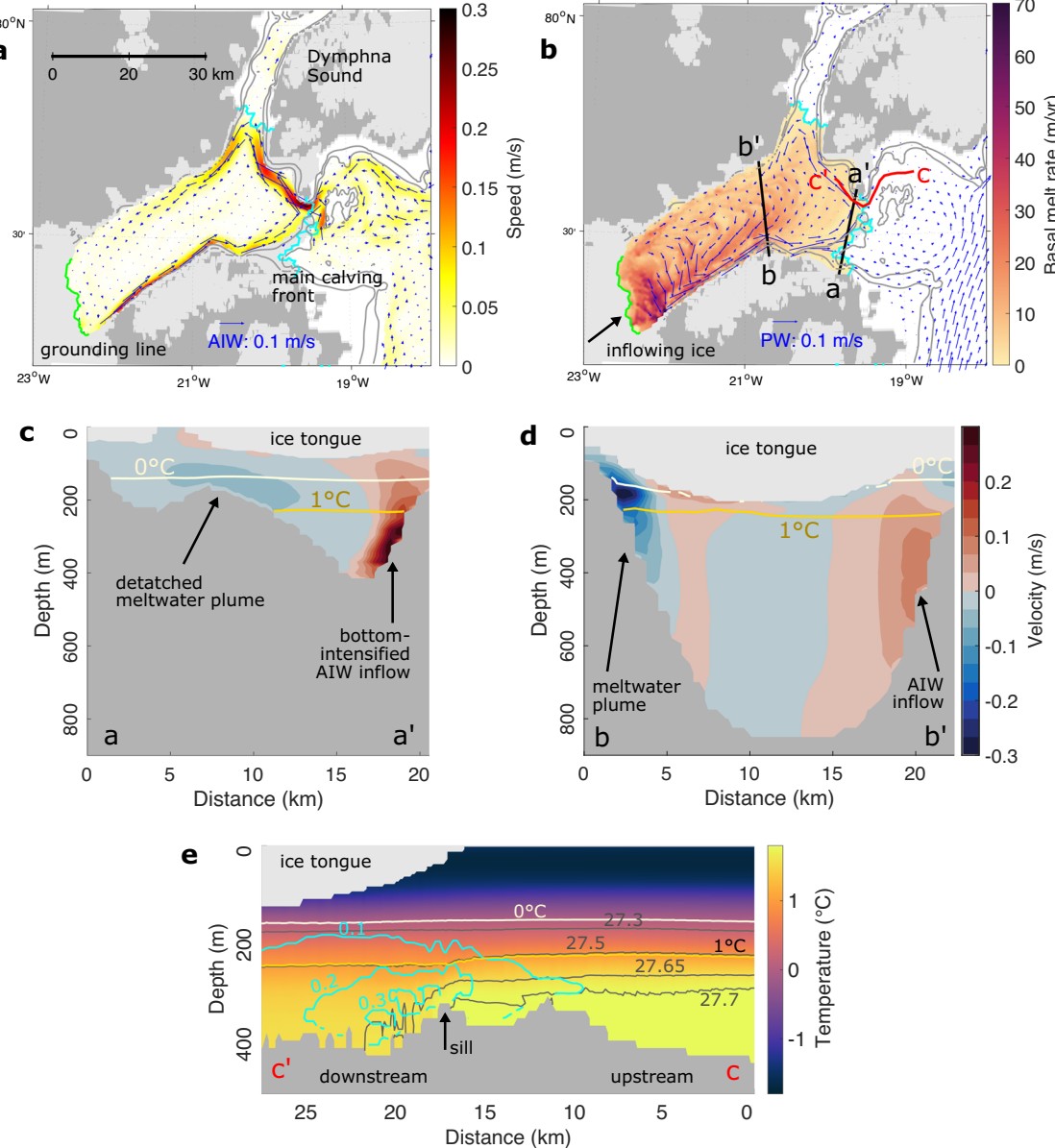

**Fig. 2 | Cavity circulation and basal melt rate. a** Simulated velocity (blue arrows and colors) in the Atlantic Intermediate Water (AIW) layer (defined by Temperature T > 1°C), and **b** melt rate at the base of the floating ice tongue (colours) and velocity in the upper water column (T < 1°C, blue arrows). Lines indicate the grounding line (green), calving front (cyan) and 100 m/200 m isobaths (gray). The gray and light gray background shows bare land and grounded ice, respectively. Cross-sectional velocity **c** close to the calving front (a–a') and **d** at the central part of the cavity (b–b'), indicated by black lines in panel **b**. 0°C and 1°C isotherms are shown. Velocity in the transects is directed positively towards the grounding line. **e** Potential temperature distribution along a transect from the continental shelf into the cavity (c–c'). At 17 km, a sill constrains the inflow of AIW. Gray contours show isopycnals, and cyan contours show the speed in m s⁻¹. The speed increases and density decreases downstream of the sill. All model results were averaged over the time period 1970–2021.

plume is strongly diluted (considering the freshwater input of 0.54 mSv and an outflow of 76 mSv, the plume contains around 1–2% meltwater).

**Trend and interannual variability of basal melt rate**
The simulated basal melt of the 79NG shows pronounced interannual variability between 1970 and 2021 (standard deviation of annual means: 3.9 km³ yr⁻¹), superimposed on a multi-decadal increase yielding a linear trend of 2.3 km³ yr⁻¹ decade⁻¹ (Fig. 3a). The trend is even stronger during the last two decades (3.4 km³ yr⁻¹ decade⁻¹). In the following we will disentangle the impacts of two main drivers of basal melt, namely ocean heat supply and subglacial discharge. Concurrent to the increase in basal melt rates, the simulated maximum

temperature of AIW at the calving front (standard deviation of 0.36°C) increased by 0.19°C decade⁻¹ during 1970–2021. This warming is consistent with the warming in the Atlantic Water layer upstream at Fram Strait (Fig. 4a, b). Besides the warming trends in the ocean, increasing air temperatures over the NEG ice sheet in the last decades[13,32] have led to higher ablation and thus increased subglacial discharge entering the 79NG cavity. The subglacial discharge rate (standard deviation of 2 km³ yr⁻¹) has more than doubled with values of 3.3 km³ yr⁻¹ in the 1970s and 6.7 km³ yr⁻¹ in the 2010s, featuring a linear trend of 0.9 km³ yr⁻¹ decade⁻¹. Given the simultaneous increase of AIW temperature and subglacial discharge, we will next quantify their relative contributions to the evolution of the basal melt of the 79NG, considering interannual and long-term variability.

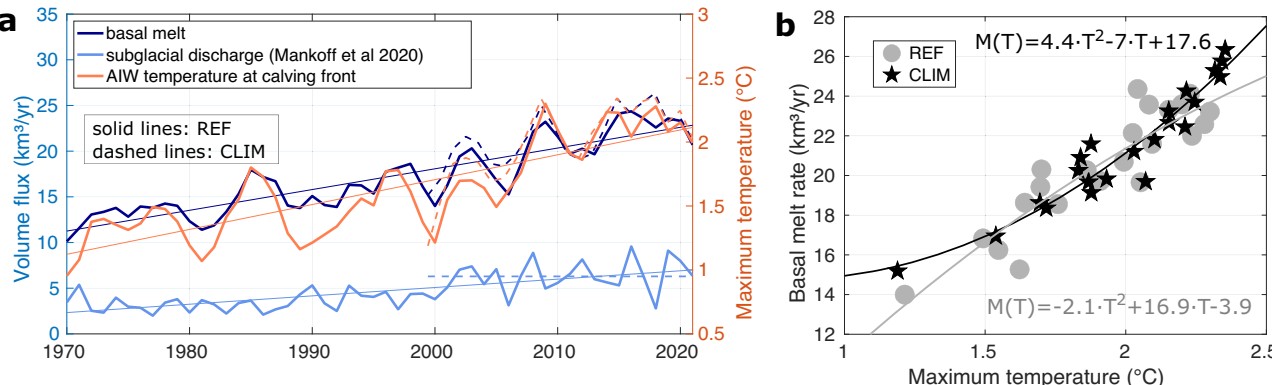

**Fig. 3 | Temporal variability. a** Annual mean basal melt rate (dark blue) and maximum Atlantic Intermediate Water (AIW) temperature in the water column at the calving front of the glacier (orange) simulated by FESOM2.1. Annual mean subglacial discharge entering the cavity at the grounding line of the glacier (light blue, ref. 54). Note that basal melt and subglacial discharge share the same y-axis. Thin colored lines show the respective linear trends of the time series for the time period 1970–2021. Solid and dashed lines correspond to the reference experiment (REF) and a sensitivity experiment with climatological subglacial discharge (CLIM), respectively. **b** Annual mean basal melt rate $M$ for years 2000–2021 as a function of maximum AIW temperature $T$ at the calving front from the REF (gray circles) and CLIM (black stars) experiments. Lines show quadratic functions fitted to the REF (gray) and CLIM (black) data points.

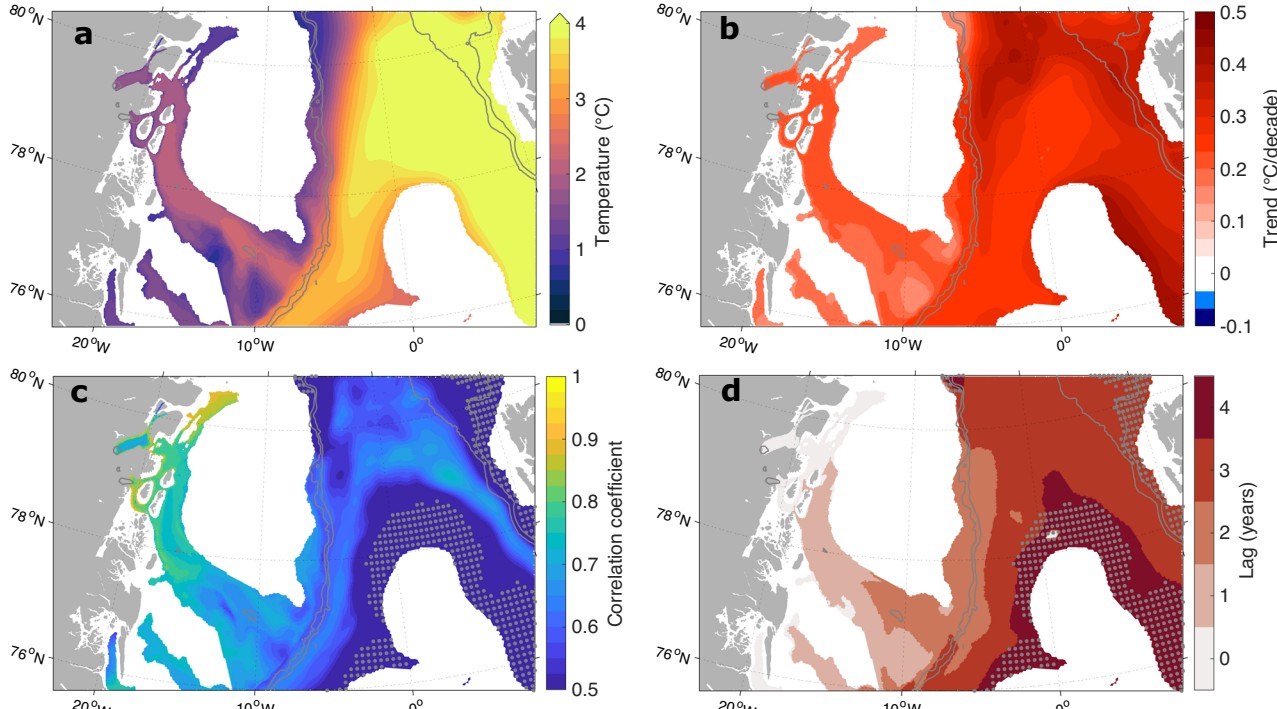

**Fig. 4 | Pathways of Atlantic Intermediate Water (AIW) towards the glacier.**
**a** Simulated mean of maximum temperature in the water column (1970–2021) and
**b** its linear trend. **c** Maximum of the lagged point-wise correlation between the annual mean vertical maximum potential temperature in each model grid point occupied by Atlantic (Intermediate) Water and basal melt rate of the 79 North Glacier and **d** the corresponding lag in years. Time series have been detrended prior to computing correlations (Methods). Gray dots indicate areas with the significance of the correlation below the 95% confidence level. Only profiles with water temperatures above 0 °C were considered. Gray contours indicate the 1000 m and 2000 m isobaths.

The annual mean basal melt of the 79NG shows a significant positive correlation with the maximum AIW temperature at the calving front for the time period 1970–2021 ($r = 0.85$, 99% significant, Fig. 3a). The latter can be traced back into Norske Trough (with a lag of one year), the continental shelf break (two years lag) and into the wider Fram Strait area (three years lag), as revealed by point-wise lagged correlations (Fig. 4c, d). The correlation map thus visualises the pathway that Atlantic Water takes before reaching the NEG glaciers. Note that a mooring array across Fram Strait has been continuously monitoring ocean temperatures in the WSC since 1997[22]; this data

source could thus be exploited to predict basal melt rates at the 79NG three years in advance. Anomalously high basal melt rates are not only associated with increased temperatures but also with an acceleration of the entire cavity circulation. Composites of cross-sectional velocity at the calving front during years of low/enhanced basal melt show a pronounced decrease/increase in the speed of both the inflow and outflow respectively (Supplementary Fig. 3).

As a side note, we find a similar high positive correlation of basal melt and AIW temperatures at ZI, 79NG's neighboring glacier ($r = 0.64$ for the time period 1970–2009, 99% significant). However, our ocean

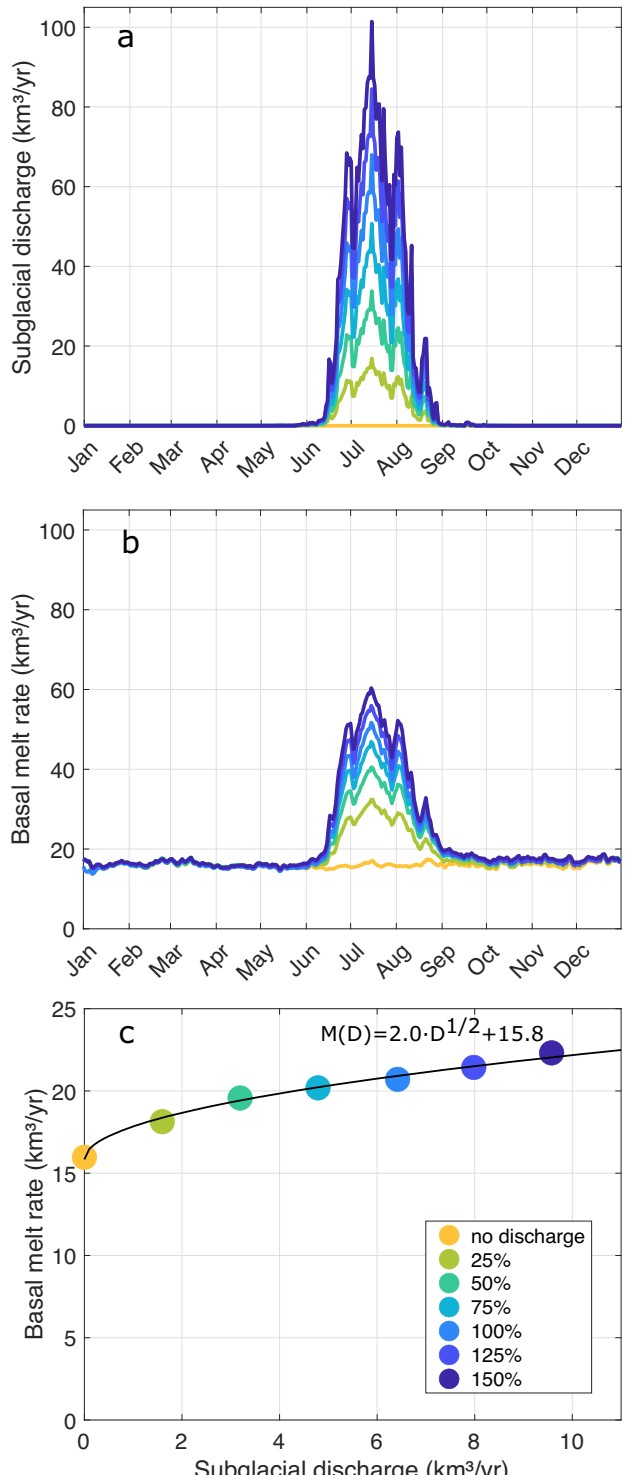

**Fig. 5 | Impact of subglacial discharge on basal melt rates of the 79 North Glacier (79NG). a** Seasonal evolution of subglacial discharge entering the cavity of the 79NG at the grounding line from[54], and **b** the simulated basal melt rate averaged over the years 2010–2014 from sensitivity experiments with varying subglacial discharge as shown in **a. c** Basal melt rate *M* as a function of subglacial discharge *D*, averaged over the time period 2010–2014, and square root function fitted to the data points (black line). In all panels, the blue color corresponds to the reference experiment (termed 100%). Yellow–green and dark blue colors correspond to sensitivity experiments with reduced (0%, 25%, 50%, 75%) or enhanced subglacial discharge (125%, 150%).

model does not resolve its transition into a tidewater glacier that started in the 2010s. An ocean model with changing ice tongue geometry would be needed here, a task left for future research.

To isolate the effect of ocean temperatures on basal melt rates, we conducted a 22-year long sensitivity experiment in which the input of interannually-varying subglacial discharge (light blue line in Fig. 3a) is replaced by a climatological seasonal cycle (experiment CLIM, Methods). The resulting basal melt rates and maximum AIW temperature at the calving front are shown in Fig. 3a (dashed lines). By eliminating the effect of interannual variability of subglacial discharge, we obtain a higher correlation of basal melt rate and ocean temperatures of $r = 0.89$ (99% significant, time period 2000–2021), compared to $r = 0.77$ (99% significant, time period 2000–2021) in experiment REF. Melt rates increase quadratically with the increase of ocean temperatures in the absence of subglacial discharge variation (black line in Fig. 3b).

The reference run does not reveal a significant correlation between basal melt and subglacial discharge ($r = 0.22$, not significant at a 5% error probability, Fig. 3a). We conducted a suite of sensitivity experiments with reduced (0%, 25%, 50%, and 75%) and enhanced (125%, 150%) subglacial discharge (Fig. 5a), branched off from the reference experiment in 2010 and run for another 5 years (see Methods). The changes in freshwater input associated with altered buoyancy lead to a response in the speed of the meltwater plume in the cavity relative to the reference run. As the basal melt rate is computed based on turbulent heat and salt fluxes with coefficients depending on the friction velocity, the melt rate responds on the order of days. A reduction/increase in subglacial discharge decreases/enhances basal melt rates (Fig. 5b). In the absence of summer subglacial discharge, basal melt rates are nearly constant throughout the year. The relationship between basal melt rate and subglacial discharge, averaged over the five years of the experiment, reveals a square root dependency (Fig. 5c).

To summarize, the basal melt rates exhibit different sensitivities to ocean temperatures and subglacial discharge. Our experiments revealed that basal melt has a quadratic dependency on AIW temperature and a square-root dependency on subglacial discharge (Figs. 3b and 5c). Similarly, model studies[19,20] and observations[33] pointed to an above linear dependency of basal melt on thermal forcing, and a square-root dependency on subglacial discharge. These relationships, derived for ice tongues and ice shelves, vary slightly from the ones derived for tidewater glaciers where subglacial discharge can exceed the subsurface ice melt. Here, a linear relationship with thermal forcing and a cubic relationship with subglacial discharge has been obtained[12,34–36]. Implications of the sensitivities found in our study for the future state of the ice tongue will be discussed now.

## Discussion

The 79NG features Greenland's largest floating ice tongue and its fate remains uncertain. Ice sheet-model experiments with varying prescribed basal melt suggest that the extent of the 79NG might not change significantly over the next century[37]. However, it has been thinning in the last decade[5,6]. A recent study based on remote sensing data detected changes in the calving front, which indicate an onset of destabilization[38]. A retreat accompanied by a reduction in buttressing is thought to increase the ice discharge upstream with implications for global sea level rise. Our inferred sensitivities may be used together with climate model future projections of subglacial discharge and AIW temperatures to predict future basal melt rates.

We demonstrated that ocean temperatures in the AIW layer dominate the interannual variability of basal melt at 79NG. CMIP6 models predict that the ocean temperature on the NEG shelf will

increase through this century[39]. In particular, the CMIP6 multi-model mean maximum temperature in the water column at the entrance of Norske Trough obtained under scenarios SSP1-2.6, SSP2-4.5, SSP3-7.0 and SSP5-8.5 increases by 1.8 °C, 2.7 °C, 3.6 °C, and 4.7 °C, respectively, between the time periods 1981–2000 and 2081–2100. Assuming that the extent of the 79NG ice tongue does not change, the quadratic dependence of basal melt on AIW temperature (Fig. 3b) suggests that the potential basal melt would be 40 km$^3$ yr$^{-1}$, 63 km$^3$ yr$^{-1}$, 92 km$^3$ yr$^{-1}$ and 140 km$^3$ yr$^{-1}$, respectively in these warming scenarios. Even in the scenario with low greenhouse gas emissions (SSP1-2.6), climate warming would lead to a 161% relative increase in basal melt rates from the time period 1981–2000 to the end of this century. In case of the high-emission scenarios, the ice tongue of the 79NG will most likely disintegrate, thus turning the 79NG into a tidewater glacier. However, in the future, we also expect that meltwater from the Greenland ice sheet increases due to increased air temperatures. In the CMIP5 RCP8.5 scenario, a 3.3-times increase in Greenland ice sheet runoff at the end of this century has been estimated[40]. Thus one expects that subglacial discharge might play an increasingly important role in the future. Given the square-root relationship obtained from our model (Fig. 5c), a 3.3-times increase in annual mean subglacial discharge would lead to a 18% increase in annual mean basal melt rates of the 79NG.

To summarize, the 79NG is unlikely to transition to a state where the atmosphere dominates the submarine melt, as is the case in northwestern Greenland[11]. In the coming decades, we suppose that the ocean might still be the dominant factor for the stability of the 79NG, even if it transitions into a tidewater glacier. We deduce this from the above cited studies[12,34–36] dealing with tidewater glaciers, that obtained a linear relationship of basal melt with ocean thermal forcing, and a below-linear relationship with subglacial discharge. Nonetheless, coupled ice sheet-ocean simulations are needed to accurately predict changes in the 79NG ice tongue in the future.

## Methods

### Ocean-ice shelf-sea ice model FESOM2.1
In this study, we employ the Finite-volumE Sea ice-Ocean Model (FESOM2.1), which solves the standard set of hydrostatic primitive equations under the Boussinesq approximation with the finite volume method[21]. FESOM2.1 is formulated on unstructured meshes, allowing for mesh refinement in areas of interest. The sea ice model operates on the same mesh as the ocean model, and its details can be found in ref. 41. FESOM2.1 performs well on a global scale[42], and the global model has been used for several regional applications such as eddy dynamics in the Arctic Ocean[43], heat transport into the Barents Sea[44] and variability of AIW on the NEG continental shelf[45].

FESOM2.1 includes an ice shelf component which resolves the 79NG and ZI cavities. The treatment of cavities closely follows their implementation in the previous version FESOM1.4[46]. The interaction between ocean and ice shelf is simulated using the three-equation system[30]. It computes melting and refreezing at the ice shelf-ocean boundary layer caused by velocity-dependent salt/heat exchanges. In our setup we set the dimensionless drag coefficient $C_d$ to $1.25 \times 10^{-3}$, which is close to the optimal value determined from sensitivity tests for simulations of the Petermann Glacier[19]. Note that the simulated basal melt rates are sensitive to the choice of the drag coefficient. Sensitivity experiment with drag coefficients of $1.875 \times 10^{-3}$ and $2.5 \times 10^{-3}$ carried out for one year revealed that the basal melt was increased by 27% and by 48% respectively relative to the reference run, whereas the variability did not change (Supplementary Fig. 4).

In the regions with coarse resolution, isoneutral tracer diffusion[47] and the Gent-McWilliams (GM)[48] eddy parameterization are applied. The GM parameterization is switched off when the horizontal mesh is finer than 30 km. The vertical mixing parametrization used here is the

turbulent kinetic energy mixing computed by the CVMix package[49]. We apply a surface salinity restoration to the PHC climatology[50] with a restoring velocity of $1.929 \times 10^{-6}$ ms$^{-1}$. The restoration helps to avoid local salinity trends that can occur in response to inaccuracies in, for example, precipitation.

### Model setup
The unstructured triangular mesh designed for this study was generated with JIGSAW-GEO[51]. The background resolution in the global oceans is set to 1° and refined to 25 km north of 40 °N. In the Arctic Ocean and Nordic Seas, the mesh is refined to 4 km, and further refined to 2.5 km on the Northeast Greenland continental shelf. In the vicinity of the 79NG, mesh resolution is set to 700 m. This resolution is appropriate to resolve the Rossby radius of deformation and thus the geostrophic circulation with at least 3 grid points (Supplementary Fig. 1c, d). For vertical discretization, 86 z-levels are used, with 5 m layer thickness in the top 100 m, and coarser layer thickness towards the ocean floor. Until 900 m water depth, which is the maximum depth below the ice tongue of the 79NG, the coarsest layer thickness is set to 25 m. Bathymetry and ice shelf topography are taken from the RTopo-2.0.4 data set[1]. In our model setup, the ice shelf topography of the 79NG represents present day conditions, and does not change with time. While this assumption is appropriate for the 79NG, where the grounding line and ice shelf topography did not change significantly in the last decades[6], it does not hold for the ZI which strongly retreated starting from the 2010s[4].

Atmospheric data from JRA55-do-v1.4.0[52] was used to force the model. The dataset has a spatial resolution of 55 km and temporal resolution of 3 hours. The JRA55-do-v1.4.0 dataset also includes global river runoff. For Greenland, we replace the river runoff by data sets of solid ice and liquid water discharge from refs. 53, 54, respectively. Solid ice discharge from the Greenland ice sheet is defined as the product of ice sheet surface velocity and thickness along an outlet glacier flux gate. As the dataset only covers the years from 1986 onwards, we use an average over the 1986–1990 period prior to 1986 in the simulation. The data set of liquid water discharge contains runoff (the sum of ablation and rainfall minus retention and refreezing) from the regional climate model RACMO, in particular ice sheet runoff and land runoff, that has been routed to ice margin outlets and coast outlets, respectively, with the topography data sets ArcticDEM and BedMachine. Both runoff data sets have daily temporal resolution and high spatial resolution (~100 m; resolving individual streams). For every data point of the Mankoff et al. data sets, the nearest model grid point around Greenland is identified and solid and liquid discharge is then spread at the ocean surface over a prescribed radius of 50 km. In the cavities, the treatment of discharge is different. Here we inject the liquid discharge (now termed subglacial discharge) evenly along the grounding line, and then spread it over a radius of 10 km. Thus, at the 79NG, we inject the liquid discharge at a depth of around 500 m, whereas it is spread at the surface model layer for all other Greenland glaciers.

A control simulation was carried out for the time period 1960–2021 (experiment REF), with a time step of 3 min. The first 10 years are considered as spin-up, and are not included in the analysis. With 2880 cpus used, a throughput of 3 model years per day was achieved on the NHR@ZIB super computer.

### Limitations of the model
There are a number of limitations of the model that will be listed here.
- In our model setup we assume a fixed geometry of the ice tongue, neglecting thereby the ice-ocean interaction. The floating ice tongue of the 79NG has been thinning in the last two decades[5,6], which is thus not represented in the model geometry.
- Subglacial discharge is implemented in the model as a line plume, i.e. it is spread evenly along the grounding line. Another

approach would be to inject the discharge as axisymmetric plume which mimics the inflow through subglacial channels, assuming a fixed outlet width. Ref. 55 pointed out that the geometry of the plume (line plume vs axisymmetric plume) has implications for entrainment and basal melting and thus the structure of the meltwater plume (line plumes leading to higher basal melt rates and entrainment than axisymmetric plumes). For the 79NG, detailed observations of subglacial discharge outlets are missing. Still, the method of injecting the discharge in the model has an impact on the basal melt rates.

- Measurement campaigns at e.g., Petermann Glacier showed that the base of the ice shelf is characterized by subglacial channels of 1–2 km width transporting glacially modified water towards the calving front[56]. Observations at Thwaites Glacier, Antarctica, show that these channels play an important role for basal melting[57]. These features are however not represented in our model, leading to possibly underestimated melt rates.

- In our model setup, we do not account for tidal motions in the cavity. Observations with an ice tethered mooring revealed that tidal currents in the cavity are weak[14], being one order of magnitude smaller than the mean inflow velocity of warm AIW.

- The model presented here has a positive temperature bias of around 0.3 °C, which can be traced back to the Nordic Seas with a warmer than observed Norwegian Atlantic Current. The warm bias can presumably be related to model deficiencies in resolving lateral ocean heat fluxes due to the fact that mesoscale eddies are not fully resolved with the applied resolution in the Nordic Seas. These lateral heat fluxes play a large role in cooling the Atlantic Water carried by the Norwegian Atlantic Slope Current northwards[58]. We acknowledge that this has an effect on the simulated mean basal melt rates which are rather on the higher side; it is probably not critical for the interannual variability and sensitivities that we find.

### Sensitivity experiments
In addition to the control simulation, we conducted several sensitivity experiments. Firstly, to test the impact of ocean temperatures on basal melt rates, we carried out a 22-year simulation (2000–2021) with a climatological seasonal cycle of subglacial discharge (constructed by averaging the subglacial discharge by ref. 54 that is injected at the grounding line of the 79NG over the same time period). This simulation is termed CLIM. Secondly, to test the impact of subglacial discharge on basal melt rates, we conducted a suite of 5-year long simulations branched off in 2010 where the subglacial discharge at 79NG was set to zero, reduced (to 25%, 50% and 75% of the original data), and enhanced (to 125%, 150% of the original data).

### Correlation coefficients
Prior to computing the correlation coefficient, all time series were linearly detrended. Correlations are regarded as significant when they are different from 0 at a 95% confidence level (two-sided hypothesis test).

### Hydrographic measurements
To assess the simulated temperature variability in Norske Trough, we use CTD measurements compiled by ref. 45. It includes processed CTD data from refs. 7, 59, 60, and from the World Data Center PANGAEA[61,62], covering the time period from 1970 to 2020. As the overall number of CTD profiles is relatively low, we computed decadal averages and the corresponding standard deviation.

### Data availability
The minimal data set of FESOM2 required to reproduce the findings of this study has been deposited in the Zenodo database under accession code https://doi.org/10.5281/zenodo.10421533[63]. Full model output is available upon request to the corresponding author.

### Code availability
The FESOM2.1 source code used in this study and mesh information has been deposited in the Zenodo database under accession code https://doi.org/10.5281/zenodo.10544236[64]. The source code is also available at: https://github.com/FESOM/fesom2. A documentation of the source code and how to run the model can be found here: https://fesom2.readthedocs.io/en/latest/index.html. It includes system requirements and an installation guide.

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

## Acknowledgements

This work was supported by the Federal Ministry of Education and Research in Germany (BMBF) through the research program GROCE2 (FKZ 03F0855A) (C.W., T.K.) and the MOSAiC polar research program (grant no. 03F0889A) (Q.W.), by the Deutsche Forschungsgemeinschaft (DFG) through the Special Priority Program (SPP)-1889 "Regional Sea Level Change and Society" (A2Green) (R.M.), by the Collaborative Research Centre TRR 181 "Energy Transfer in Atmosphere and Ocean" funded by the Deutsche Forschungsgemeinschaft (DFG, German Research Foundation) - project no. 274762653 (P.S. and S.D.) and by the Helmholtz Climate Initiative REKLIM (Regional Climate Change), a joint research project of the Helmholtz Association of German research centres (HGF) (R.T., Q.W.). The authors gratefully acknowledge the computing time granted by the Resource Allocation Board and provided on the supercomputer Lise at NHR@ZIB as part of the NHR infrastructure. The calculations for this research were conducted with computing resources under the project hbk00099. We thank Nat Wilson for providing basal melt rates of the 79NG derived from satellite imagery, and Verena Haid and Suvarchal Cheedela for technical support. The authors declare no financial conflicts of interest. We acknowledge support by the Open Access Publication Funds of Alfred-Wegener-Institut Helmholtz- Zentrum für Polar- und Meeresforschung.

## Author contributions

Conceptualization: C.W., R.M., R.T., and T.K.; Conducting model simulations and creating figures: C.W.; Model development: Q.W., P.S. and S.D.; Compiling of observational data: R.M. and W.J.v.A.; Compiling of CMIP6 data: Q.S.; Data Curation: C.W. and R.M.; Writing: C.W., R.M., W.J.v.A., Q.W., R.T., P.S., S.D., T.K.; Funding acquisition: T.K. and R.T.

## Funding

## Competing interests

The authors declare no competing interests.
