## [Peer Review File · Nature Communications]

Atlantic Water warming increases melt below Northeast Greenland's last floating ice tongueREVIEWER COMMENTS

Reviewer #1 (Remarks to the Author):

Review for "Atlantic Water warming increases melt below Northeast Greenland's last floating ice tongue"

This manuscript uses a global simulation to resolve the ocean circulation in the cavity of the 79NG with 700m resolution to explain the impact of ocean and atmosphere. The authors find that, the interannual variability and long-term trend over the past 50 years of the basal melt below the 79NG are mainly associated with changes in the temperature of the AIW. This manuscript is clearly expressed the whole story, and it is recommended to accept based on my opinion.

Suggestion:

The methods parts seem not to be the exact format with this journal.

Another thing I am very curious about is, since the authors did the FESOM2.1 among the entire northeast Greenland, and for ZI, it did not collapse until 2014, so did the author analyze the same situation for ZI before it collapsed?

Reviewer #2 (Remarks to the Author):

Comments to authors

Manuscript NCOMMS-23-49679-T "Atlantic Water warming increases melt below Northeast Greenland's last floating ice tongue" by Claudia Wekerle, Rebecca McPherson, Wilken-Jon von Appen, Qiang Wang, Ralph Timmermann, Patrick Scholz, Sergey Danilov and Torsten Kanzow

This manuscript investigates using a regional and global model the melting of the 79North glacier ice tongue and addresses the role of the ocean (i.e. warming of the Atlantic intermediate Water (AIW)) and the atmosphere (i.e. increased discharge due to the warming of the atmosphere). The authors find a dominant role of the ocean in the melting of the the 79North ice tongue.

The work used a combination of numerical model and observations both to validate the results obtained by the numerical model but also to corroborate them.

The question addressed by the authors is a long-standing question and the contribution brought by this work is relevant both for the oceanographic and glaciological community.

In general, I find the work well written, the methodology used is mostly clear, the results are clearly presented and the discussion and perspective well highlighted.

I think the manuscript is worth publishing but I have a few questions for the authors.

1) Line 47-48 I find this sentence hard to read. The concept is clear, but I would suggest rephrasing

2) Line 51-54 plumes do not form only in cavities but also at the margin of outlet glaciers. The impression here is that they form only in cavities. Please rephrase accordingly.

3) Line 61 I think the "as" should be deleted

4) The authors clearly state that one of the limitations of the numerical model is the warm bias of the AIW that can be traced back to the Norwegian current. Can the author speculate on the possible reasons of this warm bias?

5) Line 131 the Three equation system used in the model is strongly dependent on the temperature/salinity and velocity estimated by the model. Previous work cited by the authors (Xu, 2012) and others (Sciascia, 2013) have shown that high resolutions, both vertical and horizontal, are needed to correctly represent the melting at the base of the ice. The resolution here is high for a

regional model but not high to correctly capture the velocity required by three equation model. Do the author have an idea on how sensitive their model is to the horizontal resolution? Earlier regional modelling studies (Cowton 2015, <https://doi.org/10.1002/2014JC010324>) tried to overcome this issue by incorporating theoretical models of plumes into the regional numerical model. Have the authors considered this option to be confident of the melting obtained in their simulations?

6) Line 252 the drag coefficient used here is slightly higher than the usual valued used in previous work. How sensitive is the model to this choice.

7) Line 270 the geometry of the ice shelf does not change in time, but which geometry did the authors use? The one of the present days? Please clarify.

8) Line 271 Maybe it would be useful for the reader to know the resolution of the atmospheric forcing to know how it compares with the resolution of the ocean model. And especially to force the surface dynamics in the most resolved areas of the model.

9) Line 282-284 the injection of liquid discharge used in the cavities is also used for the other tidewater glaciers in Greenland? Please clarify.

10) Figure S3 panel c and d is the cross-section used here the a-a' section used in the other figures of the text? Please clarify.

Reviewer #3 (Remarks to the Author):

This paper examines how ocean conditions affect the subsurface melt on the ice tongue of the 79 North Glacier, which is the largest floating ice tongue in Greenland. For this purpose, the authors uses a global ocean-circulation model, forced by an atmospheric reanalysis dataset, and with a magnified resolution in the fjord of ice tongue and adjacent open ocean regions. The model resolves the ice cavity below the ice tongue, and subsurface ice melt is represented by boundary conditions that accounts also for effects of subglacial discharge, which stems from surface glacial melt and is forced by the atmospheric conditions. The simulation focuses on the time period 1970–2021. Additional sensitivity experiments are conducted to examine the roles of oceanic temperatures and atmospherically-driven subglacial discharge for driving basal melt on the ice tongue. The results suggest that between 1970–2021 the inter-annual variations and trend in the subsurface melt are chiefly driven by temperature changes in the Intermediate Atlantic Water off Northeast Greenland. Based on the results and climate model scenarios, a discussion is given on possible basal melt rates for 79 NG in the late part of the present century.

The paper is interesting and addresses important questions related to the dynamics of the Greenland Ice Sheet and its contribution for future sea-level rise. It is generally well written and the figures are illustrative. However, the paper reports a rather complex study in a rather short format. Therefore, I deem that a few additional clarifications are needed to convey the limitations as well as the original new contributions of this work.

Major points

The model yields a mean basal melt of $\sim 17 \text{ km}^3 \text{ yr}^{-1}$, which very close to the $\sim 18 \text{ km}^3 \text{ yr}^{-1}$ that Schaffer et al. (2020) estimated from moored observation. It would be relevant to explain and discuss any possible tuning that was required to get this remarkably close correspondence. For example are tides simulated in FESOM2.1? Or have possible tidal motions in the ice cavity been accounted for in the melt parameterization? Even if no tuning was made, it is still relevant to mention and discuss how well the model reproduces the observations and possibly why.

(around L184) Schaffer (2020) reported that on weakly-to-monthly time scales the basal melt was closely correlated with the height of the $\sim 1 \text{ }^\circ\text{C}$ isotherm close to the glacier (their rationale was that the inflow was hydraulically controlled). Is this correlation also found in the present model and are there any trends or decadal variability in the $\sim 1 \text{ }^\circ\text{C}$ isotherm height near 79 NG or at the shelf break?

Could a figure similar to S3 be used to illuminate this? The fundamental question is perhaps how closely the AIW temperature is correlated to the ~ 1 °C isotherm height.

Minor points and comments

The title is fine, but maybe "Atlantic Water warming increases melt below Greenland's largest floating ice tongue" is a good alternative.

L12: discharge -> subglacial discharge

L63: Recently, Reinert et al. (2023) published an idealised two-dimensional model study of flow and basal melt in the 79 NG cavity. This paper is relevant to cite.

L102: Writing 0.88 C gives a sense that the accuracy here is 1/100 of a degree C. I would write 0.9 C instead (i.e. one significant decimal); and the same for trends on L106

L123: It would be relevant to also cite Jakobsson et al. (2020), who reported observations on a hydraulically-controlled inflow toward Ryder Ice Tongue.

L193: In Fig 5c I recommend that you add a curve showing how the annual-mean basal melt rate responds to the changes in subglacial discharge. (Showing only JJA inflates the role of subglacial discharge, and is misleading for appreciating the annual mean melt.)

L205: It may be relevant to cite Jenkins et al. (2018), who infer a quadratic dependence of basal melt on temperature from Antarctic ice shelf observations. (My understanding is that more linear melt-temperature relationships tend to be found at tidewater glaciers – where the SDG can be much larger than the subsurface ice melt – rather than at ice tongues and shelves.)

L224: (I assume that this is inferred from the results in Fig 3b? If so it could be good to state that.) Here I suggest that you write something like: "Assuming that the extent of the 79 NG Ice Tongue does not change, the quadratic dependence of melt on AIW temperature (Fig 3b) suggests that the POTENTIAL basal melt would be

However, it seems rather unlikely that the ice tongue would remain unchanged or even exist if the basal melt would reach $\sim 140 \text{ km}^3 \text{ yr}^{-1}$, which is 20 times larger than today. Therefore, it makes sense to talk about potential melt rates.

An additional caveat, I believe, is that in projections of the potential melt rates there are implicit assumptions of the AIW height in the future: if the AIW height would decrease, the potential melt could be lower.

L234: What if the 79 NG tongue disintegrates and it becomes a tidewater glacier? Does this statement still hold?

References:

Jakobsson et al., 2020: Ryder Glacier in northwest Greenland is shielded from warm Atlantic water by a bathymetric sill. <https://doi.org/10.1038/s43247-020-00043-0>

Jenkins et al., 2018: West Antarctic Ice Sheet retreat in the Amundsen Sea driven by decadal oceanic variability, *Nature Geoscience*, VOL 11, 733–738 .

Reinert, M., Lorenz, M., Klingbeil, K., Büchmann, B., and Burchard, H. (2023). High-resolution simulations of the plume dynamics in an idealized 79°N glacier cavity using adaptive vertical coordinates. *Journal of Advances in Modeling Earth Systems*, 15, e2023MS003721. <https://doi.org/10.1029/2023MS003721>

Response to reviewers

Dear Reviewers,

On behalf of all co-authors I would like to thank you all for your effort and for taking the time to evaluate our manuscript. Your supportive and constructive comments are highly appreciated. Your suggestions will improve the quality and clarity of our work.

Please find below our answers (in black) to your questions/comments (in blue). The red color indicates text that has been added in the manuscript.

Reviewer #1

This manuscript uses a global simulation to resolve the ocean circulation in the cavity of the 79NG with 700m resolution to explain the impact of ocean and atmosphere. The authors find that, the interannual variability and long-term trend over the past 50 years of the basal melt below the 79NG are mainly associated with changes in the temperature of the AIW. This manuscript is clearly expressed the whole story, and it is recommended to accept based on my opinion.

The methods parts seem not to be the exact format with this journal.

We checked the author's guide of Nature Communications and our methods section carefully, and could not find problems with the format. However, in the first submission we did not add a proper data availability statement. We now added a link to a data repository where the necessary model output is stored to reproduce the figures. The final format of the Method section will be adjusted following the editorial comment of the journal if the paper is accepted.

Another thing I am very curious about is, since the authors did the FESOM2.1 among the entire northeast Greenland, and for ZI, it did not collapse until 2014, so did the author analyze the same situation for ZI before it collapsed?

We analyzed the processes at ZI as well. Fig. A1 shows the bathymetry and ice base topography at ZI in our FESOM2.1 simulation. Time series of maximum temperature in the water column at the deepest point close to the southern calving front (orange dot in Fig. A1) and basal melt reveal a significant correlation for the time period 1970-2009 ($r=0.64$, Fig. A2). Both time series show strong positive linear trends over the 40 years. However, since the floating ice tongue disappeared in the 2010s, we decided not to focus on the ZI. A realistic simulation would require representing the thinning and disappearing of the floating ice tongue, which was not possible with our current model setup. We agree nonetheless that it is very interesting to study the transition period of the ZI, which is thus a possible research topic for the future.

We added the following paragraph in the manuscript (lines 182-185):

“As a side note, we find a similar positive correlation of basal melt and AIW temperature at ZI, 79NG’s neighboring glacier ($r=0.64$ for the time period 1970-2009, 99% significant) before its transition into a tidewater glacier that started in the 2010s. An ocean model with changing ice tongue geometry would be needed for the transition period, a task left for future research. “

Figure A1: Water depth (left) and ice base topography (right) in the model based on data from RTopo-2.0.4 (Schaffer et al. 2019). The red line shows the calving front. The orange dot indicates the location where the maximum temperature in the water column, shown in Figure A2, has been extracted. The grey and light grey background shows bare land and grounded ice, respectively.

Figure A2: Annual mean basal melt rate of the ZI (blue) and maximum AIW temperature in the water column at the calving front of the glacier (orange) simulated by FESOM2.1. Thin colored lines show the respective linear trends of the time series for the time period 1970-2009.

Reviewer #2

This manuscript investigates using a regional and global model the melting of the 79North glacier ice tongue and addresses the role of the ocean (i.e. warming of the Atlantic intermediate Water (AIW)) and the atmosphere (i.e. increased discharge due to the warming of the atmosphere). The authors find a dominant role of the ocean in the melting of the the 79North ice tongue.

The work used a combination of numerical model and observations both to validate the results obtained by the numerical model but also to corroborate them. The question addressed by the authors is a long-standing question and the contribution brought by this work is relevant both for the oceanographic and glaciological community. In general, I find the work well written, the methodology used is mostly clear, the results are clearly presented and the discussion and perspective well highlighted. I think the manuscript is worth publishing but I have a few questions for the authors.

1) Line 47-48 I find this sentence hard to read. The concept is clear, but I would suggest rephrasing

We rephrased the sentence as follows (lines 44-45):

“Apart from the ocean which carries the warm AIW into the cavity of the 79NG, the atmosphere impacts the stability of the glacier as well via subglacial discharge (Slater et al. 2022).”

2) Line 51-54 plumes do not form only in cavities but also at the margin of outlet glaciers. The impression here is that they form only in cavities. Please rephrase accordingly.

We reformulated the sentence, and it is hopefully more clear now (lines 48-50):

“The outflowing freshwater vigorously entrains ambient waters, speeds up as a buoyant plume that develops downstream of the grounding line at the base of the ice tongue, and thereby locally increases basal melt.”

3) Line 61 I think the “as” should be deleted

Corrected.

4) The authors clearly state that one of the limitations of the numerical model is the warm bias of the AIW that can be traced back to the Norwegian current. Can the author speculate on the possible reasons of this warm bias?

The Norwegian Atlantic Current, and in particular the Norwegian Atlantic Slope Current (NwASC), transports warm Atlantic Water along the Norwegian coast towards the Arctic Ocean, which is the main source of the AIW that eventually reaches the 79NG. A recent study revealed that the cooling of the NwASC along its pathway from the Greenland Scotland Ridge up north is rather due to oceanic lateral heat transfer than air-sea heat fluxes (Huang et al. 2023). Eddies play an important role for lateral heat fluxes. Our model resolution in the Nordic Seas (4 km) can be considered as eddy-permitting, but not eddy-resolving. We speculate that higher mesh resolution would increase lateral heat fluxes in the Nordic Seas, leading to stronger heat loss as the AW travels northwards, which might reduce the warm bias. The

issue of too large AW transport into the Nordic Seas can be discarded because it is in the range of observations. We added the following sentences in the section about model limitations (lines 331-335):

“The warm bias can presumably be related to model deficiencies in resolving lateral ocean heat fluxes due to the fact that mesoscale eddies are not fully resolved with the applied resolution in the Nordic Seas. These lateral heat fluxes play a large role in cooling the Atlantic Water carried by the Norwegian Atlantic Slope Current northwards (Huang et al. 2023).”

Reference:

Huang, J., Pickart, R.S., Chen, Z. et al. Role of air-sea heat flux on the transformation of Atlantic Water encircling the Nordic Seas. Nat Commun 14, 141 (2023). <https://doi.org/10.1038/s41467-023-35889-3>

5) Line 131 the Three equation system used in the model is strongly dependent on the temperature/salinity and velocity estimated by the model. Previous work cited by the authors (Xu, 2012) and others (Sciascia, 2013) have shown that high resolutions, both vertical and horizontal, are needed to correctly represent the melting at the base of the ice. The resolution here is high for a regional model but not high to correctly capture the velocity required by three equation model. Do the author have an idea on how sensitive their model is to the horizontal resolution? Earlier regional modelling studies (Cowton 2015, <https://doi.org/10.1002/2014JC010324>) tried to overcome this issue by incorporating theoretical models of plumes into the regional numerical model. Have the authors considered this option to be confident of the melting obtained in their simulations?

Due to limitations in computing time we were not able to perform simulations with higher mesh resolution than used in our study. However, the fact that we are getting a rather good agreement with the available observational data gives us certain credence in our results. Besides, our horizontal resolution is high enough to resolve the Rossby radius of deformation (Fig. S1), so that we are able to resolve rotational effects on the currents in the cavity, which clearly play a role as revealed in Fig. 2 a, c and d. Idealised 3d plume modeling of melt processes underneath the 79NG (personal communication, Markus Reinert, IOW, Germany) shows the same signature of a thick meltwater plume at the southern margin of the ice tongue when rotational effects are incorporated, and a very thin plume in the case without rotation (Reinert et al., 2023). Thus, once the plume has been established, we expect the vertical resolution to be sufficient to model it realistically. Overall, the good model performance indicates that our model configuration (with the best resolution we can afford) is sufficient for our current purpose and we thus did not consider other modeling options. However, limited realism is expected very close to the grounding line. This is firstly due to the largely unknown spatial distribution of the injection of subglacial discharge at the grounding line (something, a plume model would not be able to fix). Secondly, we use the hydrostatic approximation which might be violated in regions where the ice tongue is particularly steep. Here a plume model might help. We did not consider incorporating a plume model in the ocean model so far, but it is definitely a great idea for future work.

6) Line 252 the drag coefficient used here is slightly higher than the usual values used in previous work. How sensitive is the model to this choice?

Thank you for this question. As far as we know, there is no good observational evidence of the value of the drag coefficient. This leaves some freedom to choose an appropriate value. Thus, the drag coefficient

is often used to tune the model to reproduce a reasonable basal melt rate. We chose a drag coefficient following Cai et al. 2017, who determined that a coefficient of $1.2 \cdot 10^{-3}$ is optimal for Petermann Glacier (NW Greenland) and best reproduces the observed basal melt. Typical values of the drag coefficient range from $1.2 \cdot 10^{-3}$ to $3.0 \cdot 10^{-3}$ (see Gwyther et al. 2015, <https://doi.org/10.1016/j.ocemod.2015.09.004>). Larger values of C_D are mostly applied for “cold” ice shelves such as the ones in Antarctica, which feature stronger friction. In contrast, smaller values are applied for “warm” ice shelves such as the Greenland ice tongues which have a smoother ice surface due to higher basal melting.

We carried out two additional sensitivity experiments with larger drag coefficients ($C_D=1.875 \cdot 10^{-3}$ and $C_D=2.5 \cdot 10^{-3}$, see Fig. A3). Indeed, the simulated basal melt rate is sensitive to the choice of the drag coefficient. The overall melt rate for the year 2000 increased by 27% with $C_D=1.875 \cdot 10^{-3}$ and by 48% with $C_D=2.5 \cdot 10^{-3}$ relative to the reference run. Changing the drag coefficient adds an offset to the time series, whereas the variability does not change.

We think it is good to show the sensitivity of the model to changes in the drag coefficient, and added the figure in the Supplementary material, and also added a short description in the methods section (lines 263-267):

„Note that the simulated basal melt rates are sensitive to the choice of the drag coefficient. Sensitivity experiments with drag coefficients of $1.875 \cdot 10^{-3}$ and $2.5 \cdot 10^{-3}$ carried out for one year revealed that the basal melt was increased by 27% and by 48% respectively relative to the reference run, whereas the variability did not change (Fig. S4). “

Figure A3: Impact of basal friction. Time series of daily mean basal melt rate in the year 2000 in experiment REF (green line, using a drag coefficient of $C_D=1.25 \cdot 10^{-3}$), and experiments with higher drag coefficient ($C_D=1.875 \cdot 10^{-3}$, turquoise line and $C_D=2.5 \cdot 10^{-3}$, blue line).

7) Line 270 the geometry of the ice shelf does not change in time, but which geometry did the authors use? The one of the present days? Please clarify.

We used the RTopo-2.0.4 dataset by Schaffer et al. 2019 to obtain the geometry of the ice tongue. Compared to the previous version RTopo-2.0.1 (Schaffer et al. 2016), it contains new bathymetry data for the Northeast Greenland continental shelf. The ice thickness data of the 79NG in RTopo-2.0.4 represents present day conditions. However, the thickness of the Zachariæ Isstrøm (ZI) ice tongue in RTopo-2.0.4 corresponds to the state before it disintegrated. In contrast to the 79NG, it thus does not represent present day conditions. We excluded the analysis of the ZI from this study. A realistic simulation of the ocean circulation in the cavity of the ZI would require a more sophisticated approach with a changing (and vanishing) ice geometry over the last decades (see answer to Reviewer 1). To clarify this, we modified and added the following sentences in the methods section (lines 284-287):

“In our model setup, the ice shelf topography of the 79NG represents present day conditions, and does not change with time. While this assumption is appropriate for the 79NG, where the grounding line and ice shelf topography did not change significantly in the last decades (Mayer et al. 2018), it does not hold for the ZI which strongly retreated starting from the 2010s (Mouginot et al. 2015).”

8) Line 271 Maybe it would be useful for the reader to know the resolution of the atmospheric forcing to know how it compares with the resolution of the ocean model. And especially to force the surface dynamics in the most resolved areas of the model.

The spatial resolution of the atmospheric forcing is 55 km, much larger than the mesh resolution close to the glaciers (700 m). We added this information in the methods section (lines 288-289):

“The dataset has a spatial resolution of 55 km and temporal resolution of 3 hours.”

Despite being relatively coarse, the JRA-55 data set is commonly used for ocean models, and FESOM2.1 produces good results with this data set. We note that atmospheric and oceanic processes have different typical spatial scales. Nonetheless, we aim to explore the impact of higher resolution atmospheric forcing data in the future.

9) Line 282-284 the injection of liquid discharge used in the cavities is also used for the other tidewater glaciers in Greenland? Please clarify.

We use the liquid discharge data from Mankoff et al. which contains data for entire Greenland, so for other tidewater glaciers as well. For 79NG, we inject the liquid discharge at the grounding line (at around 500 m depth), while for all other glaciers, the liquid discharge is injected in the surface model layer and distributed within a radius of 50 km.

We reformulated the sentences a bit (lines 299-304):

“For every data point of the Mankoff et al. data sets, the nearest model grid point around Greenland is identified and solid and liquid discharge is then spread at the ocean surface over a prescribed radius of 50 km. In the cavities, the treatment of discharge is different. Here we inject the liquid discharge (now termed subglacial discharge) evenly along the grounding line, and then spread it over a radius of 10 km. Thus, in the 79NG and ZI cavities, we inject the liquid discharge at a depth of around 500 m, whereas it is spread at the surface model layer for all other Greenland glaciers.”

10) Figure S3 panel c and d is the cross-section used here the a-a' section used in the other figures of the text? Please clarify.

Yes, the section a-a' is also shown in panels c and d. We added this information in the caption and in the figures.

Reviewer #3

This paper examines how ocean conditions affect the subsurface melt on the ice tongue of the 79 North Glacier, which is the largest floating ice tongue in Greenland. For this purpose, the authors uses a global ocean-circulation model, forced by an atmospheric reanalysis dataset, and with a magnified resolution in the fjord of ice tongue and adjacent open ocean regions. The model resolves the ice cavity below the ice tongue, and subsurface ice melt is represented by boundary conditions that accounts also for effects of subglacial discharge, which stems from surface glacial melt and is forced by the atmospheric conditions. The simulation focuses on the time period 1970–2021. Additional sensitivity experiments are conducted to examine the roles of oceanic temperatures and atmospherically-driven subglacial discharge for driving basal melt on the ice tongue. The results suggest that between 1970–2021 the inter-annual variations and trend in the subsurface melt are chiefly driven by temperature changes in the Intermediate Atlantic Water off Northeast Greenland. Based on the results and climate model scenarios, a discussion is given on possible basal melt rates for 79 NG in the late part of the present century.

The paper is interesting and addresses important questions related to the dynamics of the Greenland Ice Sheet and its contribution for future sea-level rise. It is generally well written and the figures are illustrative. However, the paper reports a rather complex study in a rather short format. Therefore, I deem that a few additional clarifications are needed to convey the limitations as well as the original new contributions of this work.

The model yields a mean basal melt of $\sim 17 \text{ km}^3 \text{ yr}^{-1}$, which very close to the $\sim 18 \text{ km}^3 \text{ yr}^{-1}$ that Schaffer et al. (2020) estimated from moored observation. It would be relevant to explain and discuss any possible tuning that was required to get this remarkably close correspondence. For example are tides simulated in FESOM2.1? Or have possible tidal motions in the ice cavity been accounted for in the melt parameterization? Even if no tuning was made, it is still relevant to mention and discuss how well the model reproduces the observations and possibly why.

For this study, we used the standard FESOM2.1 version, and did not tune it specifically. It has to be said that the model has a reasonable representation of observations, but the results are still not perfect. There is a warm bias in the AIW layer on the Northeast Greenland continental shelf, which we describe in the methods section. Reviewer #2 asked to speculate about possible reasons; among other issues there might be underestimated lateral ocean heat fluxes in the Nordic Seas.

Regarding the cavity, except for the basal drag coefficient, we use the standard setting/parameters that are used by our colleagues to simulate the ocean-ice shelf interaction in the Southern Ocean, particularly Filchner-Ronne Ice shelf. Here we benefited from their experience. Reviewer #2 pointed us to the role of the drag coefficient, and sensitivity experiments (see Figure A3, and new Figure S4 in the paper) revealed that the magnitude of basal melt strongly depends on the choice of this parameter. We chose a small parameter, following Cai et al. 2017, that is better suitable for ice shelves/tongues characterized by strong basal melting (“warm” ice shelves), such as the ones in Greenland. We added a description of the sensitivity experiments with varying drag coefficients in the Methods section. We are very glad that the simulated basal melt fits well to the estimate from mooring observations by Schaffer et al. 2020, but it has to be noted again that the magnitude is sensitive to the choice of the drag parameter.

In our model setup, we do not simulate tides, and we also do not account for tidal motions in the melt parameterization. We added this model deficiency to the “Limitations of the model” section (lines 327-329):

“(4) In our model setup, we do not account for tidal motions in the cavity., Observations with an ice tethered mooring revealed that tidal currents in the cavity are weak (Lindeman et al. 2020), being one order of magnitude smaller than the mean inflow velocity of warm AIW.”

(around L184) Schaffer (2020) reported that on weakly-to-monthly time scales the basal melt was closely correlated with the height of the ~ 1 °C isotherm close to the glacier (their rationale was that the inflow was hydraulically controlled). Is this correlation also found in the present model and are there any trends or decadal variability in the ~ 1 °C isotherm height near 79 NG or at the shelf break? Could a figure similar to S3 be used to illuminate this? The fundamental question is perhaps how closely the AIW temperature is correlated to the ~ 1 °C isotherm height.

We computed the depth of the 1°C isotherm at the calving front of the 79NG, and found, similar to Schaffer et al. (2020), a high correlation with the maximum temperature in the water column ($r=0.96$, Figure A4). The depth of the 1°C isotherm also reveals a strong positive linear trend during the time period 1970-2021 (21 m/decade). The vertical displacement of the 1°C isotherm is thus connected to changes in the maximum temperature in the water column.

Figure A4: Annual mean depth of the 1°C isotherm (dark blue) and maximum AIW temperature in the water column (orange) at the calving front of the 79NG simulated by FESOM2.1. Thin colored lines show the respective linear trends of the time series for the time period of 1970-2021.

The title is fine, but maybe "Atlantic Water warming increases melt below Greenland's largest floating ice tongue" is a good alternative.

Thanks a lot for this suggestion! However, we thought to stay with the original title because it conveys the geographical region (Northeast Greenland).

L12: discharge -> subglacial discharge

Corrected. Note that we had to shorten the abstract due to the word limit (max 150 words allowed).

L63: Recently, Reinert et al. (2023) published an idealised two-dimensional model study of flow and basal melt in the 79 NG cavity. This paper is relevant to cite.

The paper by Reinert et al. is indeed very relevant to cite. We added the citation here (lines 61-63):

"Note that modelling studies exploring the ocean circulation at the 79NG and in other cavities of Greenland glaciers (e.g. Petermann and Ryder Glaciers) have been so far restricted to idealized setups in two dimensions (Reinert et al. 2023, Cai et al. 2017, Wiskandt et al. 2023)."

L102: Writing 0.88 C gives a sense that the accuracy here is 1/100 of a degree C. I would write 0.9 C instead (i.e. one significant decimal); and the same for trends on L106

Corrected.

L123: It would be relevant to also cite Jakobsson et al. (2020), who reported observations on a hydraulically-controlled inflow toward Ryder Ice Tongue.

Thanks for pointing us to the study by Jakobsson et al. (2020). We added this reference (lines 120-121):

“This is comparable to Ryder Glacier in North Greenland, which is shielded by a bathymetric sill as well (Jakobsson et al. (2020).”

L193: In Fig 5c I recommend that you add a curve showing how the annual-mean basal melt rate responds to the changes in subglacial discharge. (Showing only JJA inflates the role of subglacial discharge, and is misleading for appreciating the annual mean melt.)

We agree with the reviewer that it makes much more sense to show the annual mean basal melt rate and subglacial discharge in Figure 5c, since the focus of this study is on interannual variability. We thus decided to show only the annual means, instead of adding another curve.

This changes slightly a statement in the Discussion section, where we applied the square-root relationship between basal melt and subglacial discharge to infer the sensitivity to a potential future subglacial discharge. We thus modified the corresponding sentence in the Discussion section (lines 239-241):

*“Given the square-root relationship obtained by our model (Fig. 5c), a 3.3-times increase in **annual mean** subglacial discharge would lead to a **18% increase in annual mean** basal melt rates of the 79NG.”*

L205: It may be relevant to cite Jenkins et al. (2018), who infer a quadratic dependence of basal melt on temperature from Antarctic ice shelf observations. (My understanding is that more linear melt–temperature relationships tend to be found at tidewater glaciers – where the SDG can be much larger than the subsurface ice melt – rather than at ice tongues and shelves.)

We cited the study by Jenkins et al. (2018), and also added a relevant paper by Slater et al (2018). Thank you for clarifying the difference between ice tongues/shelves and tidewater glaciers. We added this distinction in the manuscript (lines 208-213):

*“Similarly, model studies (Wiskandt et al. 2023, Cai et al. 2017) and observations (Jenkins et al. 2018) pointed to **an above-linear dependency** of basal melt on thermal forcing, and a square-root dependency on subglacial discharge. **These relationships, derived for ice tongues and ice shelves, differ slightly from the ones derived for tidewater glaciers where subglacial discharge can exceed the subsurface ice melt. In the latter case, a linear relationship with thermal forcing and a cubic relationship with subglacial discharge have been obtained (Xu et al. 2012, Slater et al. 2016, Cenedese et al. 2016, Jackson et al. 2022).”***

L224: (I assume that this is inferred from the results in Fig 3b? If so it could be good to state that.) Here I suggest that you write something like: "Assuming that the extent of the 79 NG Ice Tongue does not change, the quadratic dependence of melt on AIW temperature (Fig 3b) suggests that the POTENTIAL basal melt would be

However, it seems rather unlikely that the ice tongue would remain unchanged or even exist if the basal melt would reach $\sim 140 \text{ km}^3 \text{ yr}^{-1}$, which is 20 times larger than today. Therefore, it make sense to talk about potential melt rates. An additional caveat, I believe, is that in projections of the potential melt rates there are implicit assumptions of the AIW height in the future: if the AIW height would decrease, the potential melt could be lower.

Thanks a lot for suggesting a much better sentence. We modified it accordingly (lines 229-232):

“Assuming that the extent of the 79NG ice tongue does not change, the quadratic dependence of basal melt on AIW temperature (Fig. 3b) suggests that the potential basal melt would be 40 km³yr⁻¹, 63 km³yr⁻¹, 92 km³yr⁻¹ and 140 km³yr⁻¹, respectively in these warming scenarios.”

Of course it makes much more sense to talk about potential melt rates. This whole paragraph is speculative, and certainly our discussion is not taking into account any dynamic response of the ice tongue.

Regarding the height of the AIW layer, we showed above (following your suggestion), that the maximum temperature of the AIW layer is highly correlated with the AIW layer thickness (represented by the 1°C isotherm). This correlation might of course break down in the future.

L234: What if the 79 NG tongue disintegrates and it become a tidewater glacier? Does this statement still hold?

We suppose that this statement still holds if the 79NG transitions into a tidewater glacier. The studies cited above (Xu et al. 2012, Slater et al. 2016, Cenedese et al. 2016, Jackson et al. 2022) dealing with tidewater glaciers still find a linear relationship of basal melt with ocean thermal forcing, and a below-linear relationship with subglacial discharge. To study the possible future shape of the 79NG and its sensitivity to ocean temperatures and subglacial discharge, a coupled ocean-ice sheet is needed though. We modified the last paragraph slightly (lines 243-247):

“In the coming decades, we suppose that the ocean might still be the dominant factor for the stability of the 79NG, even if it transitions into a tidewater glacier. We deduce this from the above cited studies (Xu et al. 2012, Slater et al. 2016, Cenedese et al. 2016, Jackson et al. 2022) dealing with tidewater glaciers, that obtained a linear relationship of basal melt with ocean thermal forcing, and a below-linear relationship with subglacial discharge.”

References:

Jakobsson et al., 2020: Ryder Glacier in northwest Greenland is shielded from warm Atlantic water by a bathymetric sill. <https://doi.org/10.1038/s43247-020-00043-0>

Jenkins et al., 2018: West Antarctic Ice Sheet retreat in the Amundsen Sea driven by decadal oceanic variability, Nature Geoscience, VOL 11, 733–738 .

Reinert, M., Lorenz, M., Klingbeil, K., Büchmann, B., and Burchard, H. (2023). High-resolution simulations of the plume dynamics in an idealized 79°N glacier cavity using adaptive vertical coordinates. Journal of Advances in Modeling Earth Systems, 15, e2023MS003721. <https://doi.org/10.1029/2023MS003721>

REVIEWERS' COMMENTS

Reviewer #2 (Remarks to the Author):

I thank the authors for addressing carefully all the comments of the reviewers and modifying the manuscript accordingly.

I think the manuscript is now suitable for publication.

Reviewer #3 (Remarks to the Author):

The authors have addressed all my comments constructively. I recommend that the paper is published in its present form.